# Demographic-, Radiographic-, and Surgery-Related Factors Do Not Affect Functional Internal Rotation Following Reverse Total Shoulder Arthroplasty: A Retrospective Comparative Study

**DOI:** 10.3390/healthcare12171695

**Published:** 2024-08-26

**Authors:** Felix Hochberger, Jakob Siebler, Marco-Christopher Rupp, Bastian Scheiderer, Sebastian Siebenlist, Stephanie Geyer

**Affiliations:** Department of Orthopaedic Sports Medicine, Technical University of Munich, Ismaninger Str. 22, 81675 Munich, Germany; felix.hochberger@klh.de (F.H.); marco-christopher.rupp@tum.de (M.-C.R.); bastian.scheiderer@tum.de (B.S.); stephanie.geyer@tum.de (S.G.)

**Keywords:** reverse total shoulder arthroplasty, internal rotation, scapular notching, complications, radiologic parameters

## Abstract

Purpose: This study aimed to identify the demographic-, radiographic-, and surgery-related factors influencing postoperative functional internal rotation (fIR) following reverse total shoulder arthroplasty (RTSA). Methods: In this retrospective cohort study, patients who underwent RTSA between June 2013 and April 2018 at a single institution were assigned to two groups (“IROgood” or “IRObad”). Patients were classified as having good fIR (≥8 points in the Constant–Murley score (CS) and fIR to the twelfth thoracic vertebra or higher) or poor fIR (≤2 points in the CS and fIR to the twelfth thoracic vertebra or lower) after RTSA with a single implant model. The minimum follow-up period was two years. Standardized shoulder-specific scores (Visual Analogue Scale (VAS), Simple Shoulder Test (SST), American Shoulder and Elbow Surgeons Score (ASES), Constant–Murley score (CS)) were used to assess the pre- and postoperative functional status of patients. Postoperative radiographic evaluation included the distalization shoulder angle (DSA), lateralization shoulder angle (LSA), critical shoulder angle (CSA), acromiohumeral distance (AHD), glenoid inclination (GI), medialization of the center of rotation (COR), lateralization of the humerus, and distalization of the greater tuberosity. Additionally, preoperative evaluation included rotator cuff arthropathy according to Hamada, glenoid version, anterior or posterior humeral head subluxation, and fatty infiltration of the rotator cuff according to Goutallier. Univariate analysis of demographic, surgical, radiographic, and implant-associated parameters was performed to identify factors associated with postoperative fIR. The Shapiro–Wilk test assessed the normal distribution of the data. Intergroup comparisons regarding demographic and surgery-related factors were conducted using the Mann–Whitney-U Test. Radiographic changes were compared using chi-square or Fisher’s exact tests. The significance level was set at *p* < 0.05. Results: Of a total of 42 patients, 17 (age: 73.7 ± 5.0 years, follow-up (FU) 38 months [IQR 29.5–57.5]) were included in the “IRObad” group, and 25 (age: 72 ± 6.1 years, FU 47 months [IQR 30.5–65.5]) were included in the “IROgood” group. All patients were treated with the same type of implant (glenosphere size: 36 mm, 14.3%; 39 mm, 38.1%; 42 mm, 47.6%; neck-shaft angle: 135° in 68.0%; 155° in 32.0%) and had comparable indications. Univariate analysis did not reveal any of the investigated demographic, radiographic, or surgery-related parameters as risk factors for poor postoperative fIR (*p* > 0.05). Conclusion: None of the investigated factors, including implant-associated parameters, influenced postoperative fIR after RTSA in this cohort.

## 1. Introduction

Reverse shoulder arthroplasty (RTSA), developed by Paul Grammont in 1985, has steadily gained popularity as a treatment for various shoulder conditions [1]. Originally used for rotator cuff arthropathy, clinical indications have expanded to include omarthrosis, proximal humeral fractures in the elderly, neoplasms, refractory glenohumeral instability, irreparable rotator cuff tears, and pseudoparalysis [2,3]. The initial principle of RTSA is based on a distalization and medialization of the center of rotation and thus a restoration of function despite a defective rotator cuff due to restoration of tension of a shortened deltoid muscle and enhancement of the efficiency of the muscle by approximately 30% [4,5]. Various modifications to the original Grammont design have been proposed to reduce the risk of failure and complications, as well as to enhance clinical outcomes [1]. RTSA has proven to be an effective treatment for reducing pain and improving shoulder function and thus has been shown to be an adequate way to restore activity of daily living [1]. According to the current literature, 10-year-survival rates are estimated to be over 90% [6,7], with an overall complication rate of approximately 9.4% [8]. If complications arise, besides scapular notching, glenoid loosening, and infection, limitations in external and internal rotation are frequently observed [1]. Over the years, ongoing improvements of implants, surgical techniques, and rehabilitation protocols have been aimed at further reducing these complications and enhancing clinical and functional outcomes. Therefore, RTSA has been a major topic of orthopedic research in recent decades. A growing body of biomechanical and clinical evidence has demonstrated that improvements in abduction and flexion can be expected following RTSA [7,9,10,11]. However, fIR of the shoulder often remains insufficient, potentially leading to significant limitations in daily activities [12,13]. Previous studies have analyzed various surgical and implant-related factors affecting fIR. Early biomechanical studies suggested that several factors are positively associated with increased fIR after RTSA [14,15,16]. These factors include lateralization of the center of rotation (COR) [17,18,19], inferior positioning of the baseplate [20,21], smaller glenosphere [22], reduced thickness of the humeral insert [23,24], neck-shaft angle (NSA) less than 155° [25], an intact subscapularis (SSC) [26] and humeral retroversion < 20 degrees. However, most of these findings have only been partially confirmed in clinical studies to date, or they present contradictory results [27,28,29,30,31]. For instance, glenoid lateralization has been reported to positively impact postoperative fIR [28,32], thereby corroborating the biomechanical results of Keener and Liou et al. [17,19]. Conversely, a recent study by Clinker et al. suggested that reduced lateralization is associated with improved functional outcomes after two years [29]. Furthermore, the choice of neck-shaft angle (NSA) has yielded mixed results. Two recent studies by Macken and Neyton et al. reported differing conclusions: one demonstrated that an NSA of 155° without additional subscapularis (SSC) refixation leads to favorable clinical and functional outcomes, including fIR [30], while the other indicated that an NSA of 135° compared to 145° does not result in significant differences in clinical and functional outcomes, though it shows a lower propensity for scapular notching [33]. Regarding the impact of SSC refixation on fIR, recent studies predominantly support the idea of tendon refixation, with findings suggesting its beneficial effects on postoperative fIR [27,28]. Overall, based on these—in parts conflicting—findings, partly due to the lack of standardized study designs and lack of inclusion of preoperative data and the use of different prosthetic components, there is still a gap in knowledge and consensus about which surgery-related, radiographic, and implant-associated factors should be carefully considered when attempting to ensure postoperative fIR after RTSA.

Therefore, the objective of the present study was to identify potential factors that might be related to good or poor fIR in patients treated with RTSA with non-uniform prosthetic components who had overall subjectively satisfactory and comparable functional outcomes postoperatively. We hypothesize that achieving a satisfactory fIR after RTSA depends on multiple patient-specific factors and less on individual implant-associated or radiological factors.

## 2. Materials and Methods

### 2.1. Study Population

This retrospective cohort study evaluated patients who underwent reverse total shoulder arthroplasty (RTSA) at a single institution from June 2013 to April 2018, with a minimum follow-up period of two years. Prior to initiating the study, institutional review board approval was secured (IRB No. 269/19s on 19 September 2019). Additionally, approval for obtaining postoperative X-rays was granted by the Federal Office for Radiation Protection (No. Z 5-22464/2019-121-A). All participants provided informed consent. Some patients included in this study were also part of previous research addressing unrelated questions. The study’s inclusion criteria encompassed individuals with rotator cuff arthropathy, primary osteoarthritis, posttraumatic arthritis, and acute humeral head fractures. Exclusion criteria included patients with a history of ipsilateral shoulder arthroplasty, vascular diseases (to mitigate risks related to blood flow and healing), malignant conditions (to prevent interference with overall health and recovery), or dementia (due to difficulties in adhering to postoperative care and rehabilitation). The presence or absence of subscapularis repair was not a criterion for inclusion or exclusion but was assessed as a potential factor influencing functional internal rotation (fIR).

### 2.2. Surgical Intervention and Postoperative Rehabilitation Protocol

All surgeries were performed by a single fellowship-trained orthopedic shoulder surgeon using a standardized technique. The Universe Reverse (Arthrex Inc., Naples, FL, USA) prosthesis, a 135° inlay-types modular implant system, was implanted in all cases. The arthroplasty procedure was performed according to the implant manufacturers recommendation and based on the preoperative planning using the mediCAD Hectec GmbH planning software (Altdorf/Landshut, Germany). The deltopectoral approach was utilized in every procedure. Surgical access to the glenohumeral joint was achieved through an SSC release using the peel-off technique. Repair of the subscapularis (SSC) was attempted in all patients, if possible, otherwise a tenotomy was performed. The humeral component was placed with an intramedullary guide, set at 20° of retroversion, tailored to the patient’s anatomy. The glenoid baseplate was positioned at the inferior edge of the glenoid. Generally, a size 42 glenosphere was used to ensure adequate inferior overhang, though a size 38 glenosphere was occasionally employed for small female patients.

All patients followed a standardized rehabilitation protocol regardless of the type of intraoperative component selection and SSC refixation. After surgery, patients’ shoulders were immobilized in an abduction brace for a period of four weeks, accompanied by active-assisted mobilization. At the five-week mark, patients advanced to active range of motion and strengthening exercises. Return to sports activities was allowed four months following the procedure.

### 2.3. Clinical Outcome Measures

Preoperative and postoperative clinical evaluations were carried out by independent shoulder surgeons from the same department, encompassing a range of experience levels from non-specialists to highly experienced specialists, none of whom were involved in the surgical procedures. These assessments involved measuring both active and passive range of motion (ROM) with a handheld goniometer, as well as evaluating the Constant–Murley score (CS), including both unadjusted and age- and sex-adjusted versions [34,35], which were obtained at least two years after the surgery. Active shoulder range of motion (ROM) was assessed with the patient standing using a goniometer. Abduction strength was evaluated using an isometric dynamometer (Isobex, Cursor AG, Bern, Switzerland), following the manufacturer’s specifications.

### 2.4. Radiographic Assessment

Preoperative and postoperative radiographic evaluations included true antero-posterior, axial, and y-view images, which were also obtained during follow-up visits. Additionally, preoperative assessment involved computed tomography (CT) scans. Radiographic measurements on preoperative X-rays encompassed the distalization shoulder angle (DSA), lateralization shoulder angle (LSA) [36], critical shoulder angle (CSA) [37], acromiohumeral distance (AHD), glenoid inclination (GI) [38], medialization of the center of rotation (COR), lateralization of the humerus, distalization of the greater tuberosity, and assessment of rotator cuff arthropathy according to Hamada. CT scans were used to determine glenoid version, anterior or posterior humeral head subluxation, and fatty infiltration of the rotator cuff according to Goutallier [39]. AHD was measured as the perpendicular distance from the most lateral part of the acromion’s undersurface to a line parallel to the superior border of the greater tuberosity [40]. LSA was defined as the angle between a line extending from the superior glenoid tubercle to the lateral border of the acromion and a line from the acromion’s lateral border to the lateral border of the greater tuberosity [36]. DSA was calculated between a line from the acromion’s lateral border to the superior glenoid tubercle and another from the tubercle to the superior border of the greater tuberosity [36]. CSA was determined by the angle between a line from the glenoid’s superior pole to its inferior pole and a line from the inferior pole to the acromion’s lateral edge [37] (Figure 1). Postoperative measurements included DSA, LSA, medialization of the COR, lateralization of the humerus, distalization of the greater tuberosity, baseplate inclination (BI), glenoid overhang (GO), and inferior scapular notching (SN) according to Sirveaux [41] (Figure 2). Glenoid inclination was measured as the angle between the floor of the supraspinatus fossa and the glenoid fossa, while baseplate inclination was the angle between the supraspinatus fossa and the back of the glenoid component [38] (Figure 1).

Preoperative and postoperative radiographic assessments were conducted by two independent observers: a highly experienced specialist (BS) and a non-specialist (FH). The final results were established through consensus between the observers.

### 2.5. Group Allocation

Patients were categorized into groups based on their postoperative functional internal rotation (fIR) following the criteria set by Hochreiter et al. [42]. The “IROgood” group included patients with good fIR, defined as internal rotation reaching the twelfth thoracic vertebra or higher and a Constant–Murley score (CS) of 8 or more. Conversely, the “IRObad” group consisted of patients with poor fIR, characterized by internal rotation to the twelfth thoracic vertebra or lower and a CS of 2 or less. Group allocation is displayed in Figure 3. Only patients meeting both criteria for a group were included in the evaluation; those meeting only one criterion were excluded.

### 2.6. Univariate Comparison

Univariate statistical analysis was conducted for each group separately to assess differences in demographic, radiographic, and implant-associated parameters between patients with favorable postoperative functional internal rotation (“IROgood” group) and those with poor functional internal rotation (“IRObad” group), as defined by the criteria mentioned.

### 2.7. Statistical Analysis

All statistical analyses were performed using SPSS Statistics, version 26 (IBM Corp., Armonk, NY, USA). Continuous variables are presented as mean, minimum, maximum, and standard deviations, while categorical variables are provided as frequencies and percentages. The Shapiro–Wilk test was used to assess the normality of the data distribution. Since the data did not follow a normal distribution, the Mann–Whitney-U Test was chosen for comparing continuous variables between groups with poor (≤2 points in Constant–Murley score (CS)) and good (≥8 points in CS) functional internal rotation (fIR) after RTSA, such as demographic and surgery-related parameters (age, follow-up, BMI, ASA, etc.). This test is appropriate for non-parametric data and does not assume normal distribution. Radiographic changes assessed with the Mann–Whitney-U Test included DSA, LSA, CSA, AHD, GI, COR medialization, humerus lateralization, greater tuberosity distalization, glenoid version, glenoid inclination, BI, and GO. To compare categorical variables, the chi-square test or Fisher’s exact test was used, depending on the expected frequencies. These tests were chosen because they are suitable for analyzing the relationships between categorical variables and do not assume normal distribution. Variables analyzed included humeral head subluxation, rotator cuff arthropathy (Hamada), and fatty infiltration (Goutallier, SN). The significance level was set at *p* < 0.05.

## 3. Results

### 3.1. Baseline Characteristics

An analysis of the institutional database identified 68 patients who underwent reverse total shoulder arthroplasty (RTSA) between June 2013 and April 2018. Of these, four patients were excluded due to a prior arthroplasty of the ipsilateral shoulder, six due to vascular or malignant diseases, and two due to dementia. Additionally, five patients were excluded due to incomplete data, two declined participation, and seven did not meet the group allocation criteria. This left 42 patients who met the inclusion criteria for final evaluation (Figure 4).

Among these, 17 patients were classified into the “IRObad” group and 25 into the “IROgood” group. Glenosphere diameters were 36 mm for 6 patients (14.3%), 39 mm for 16 patients (38.1%), and 42 mm for 20 patients (47.6%). Glenosphere offset was standard (+0 mm lateral) for 39 patients (92.9%), lateral (+4 mm lateral) for 2 patients (4.8%), and inferior (+2.5 mm inferior) for 1 patient (2.4%). Structural bone-graft augmentation for severe glenoid bone loss was used in 7 patients (11.5%). The humeral neck-shaft angle was 135° in 29 patients (68.0%) and 155° in 13 patients (32.0%). All but 5 humeral stems (8.2%) were uncemented. Subscapularis tendon repair was performed in 31 patients (73.8%). There were no significant differences between the groups in terms of age, sex, body mass index (BMI), ASA (American Society of Anesthesiologists) score, or follow-up duration (*p* > 0.05). Baseline characteristics of the two groups are summarized in Table 1.

### 3.2. Functional Outcomes

Overall, no significant differences were observed between the two groups in functional postoperative status, including active and passive range of motion (ROM) and abduction strength, other than in functional internal rotation (fIR) (*p* > 0.05) (Table 2).

### 3.3. Radiographic Outcomes

Univariate analysis did not identify any of the following parameters as risk factors for poor postoperative functional internal rotation (fIR): gender, BMI, subscapularis repair, preoperative humeral head subluxation, glenoid version, or the presence of scapular notching or spur formation (*p* > 0.05). Additionally, technique-specific variables, including distalization and medialization of the center of rotation, baseplate inclination, glenoid overhang, and glenosphere size, did not significantly affect internal rotation after reverse shoulder arthroplasty (*p* > 0.05) (Table 3).

## 4. Discussion

The main finding of this study is that none of the investigated factors—demographic, radiographic, or implant-associated—significantly impacted fIR after RTSA. When comparing baseline demographics, no significant differences were observed between the two groups. However, the “IRObad” group had a higher proportion of men and individuals with a higher BMI compared to the “IROgood” group. These results are consistent with existing literature, where BMI and male gender have been associated with worse postoperative fIR following RTSA [13,26,43].

Various surgical and implant-associated parameters were investigated for their potential role in influencing postoperative fIR. In our patient population, the size of the glenosphere neither negatively nor positively affected fIR. These findings are consistent with Müller et al. [44], who, in their retrospective analysis of 68 patients treated with RTSA, showed that an increase in glenosphere diameter led to a moderate but significant increase in external rotation, adduction, and abduction strength, but had no influence on internal rotation [44]. However, the literature presents inconsistent data regarding the relationship between fIR and glenosphere size. Two independent studies have demonstrated that increasing the glenosphere size to 42 mm may improve fIR [21,45]. Conversely, other authors have shown an inverse relationship between glenosphere diameter and fIR, indicating that a larger glenosphere diameter may result in worse fIR [5,22]. Regarding distalization, our results differ from the existing literature. In their retrospective comparative study in 2021, Hochreiter et al. described a positive correlation between distalization, a high humeral inlay, and postoperative fIR [42]. However, they could not provide an explanation for this observation.

Several studies have now been published on the influence of lateralization of the glenosphere on fIR. The research groups led by Boileau et al. and Werner et al. recently demonstrated a positive impact on fIR in their retrospective clinical studies [28,32]. In contrast, an also recently published study by Clinker et al. from May 2024 suggests that a reduced lateralization is associated with better function and less pain after 2 years [29]. However, the majority of the available studies on this topic are still biomechanical studies [17,18,23], of which it is already known that changing a single variable does not always translate to clinical improvement in fIR [13]. This is most likely due to a variety of variables that act synergistically and are difficult to model. In the present study, no positive effect of lateralization on fIR was found. Moreover, the different humeral neck-shaft angles (155°/135°) could not be shown to have a positive or negative effect on postoperative fIR. Given the previous studies, these results are of interest. A review of the current literature reveals inconsistent statements in this regard. The study group led by Newton et al. have recently published the first retrospective clinical–radiological study comparing RTSA with either 145° or 135° NSA at 2-year follow-up [33]. In particular, they were able to show that the rate of scapular notching was significantly lower in RTSA with NSA of 135° compared to 145°, while comparable clinical and functional results were achieved without fIR being affected. In contrast, Macken et al. recently showed, from their 2-year follow-up, that NSA of 155° leads to a satisfactory fIR without increased rates of instability [28], even without performing additional refixation of the SSC tendon in any of the RTSA cases examined. A previous systematic review by Longo et al. from 2022 examined the available literature on the differences between NSA of <155° and 155°. Disparities were observed in terms of external rotation, revision rates, and scapular notching, but not in terms of fIR [46]. The results of the study presented here are in line with the current state of knowledge in the literature and do not allow any conclusions to be drawn about the influence of the NSA on the fIR.

The relevance of SSC repair for fIR after RTSA has been debated controversially previously. Macken et al. recently showed that a 155° implant without additional SSC repair could lead to satisfactory fIR after 2 years [30]. Conversely, the research group led by Chelli and Boileau reported that SSC repair, together with a lateralized prosthesis design, leads to improved fIR [28]. A Korean study group led by Rhee et al. recently described the significant role attributed to the preoperative muscle quality of the SSC. In their collective, they demonstrated that a lower fatty muscle infiltration was associated with an improved fIR, but also with an overall increased postoperative complication rate [31]. A recently published prospective study by a German research group led by Ameciane et al. reported improved fIR values after SSC repair 2 years postoperatively compared to those without SSC repair [27]. In the present study, we did not find a positive effect of SSC refixation on fIR. These results align with recent clinical studies that also report no significant benefit from additional SSC repair. Despite this, SSC refixation is still frequently performed in clinical practice when the tendon tissue is in good condition. Hochreiter et al. hypothesized that fIR is not primarily determined by SSC repair, but might also be influenced by other muscles such as the deltoid, pectoralis major, latissimus dorsi, or teres major [42]. No significant effect on postoperative fIR was observed for the radiographic parameters investigated in this study, which had comparable preoperative data. The effect of these parameters on fIR is controversial. A recent systematic review analyzed the existing literature on this subject and found a scarcity of evidence, resulting in occasionally controversial findings [13]. For instance, Rol et al., in their retrospective comparative clinical study, demonstrated that good preoperative fIR, a small preoperative glenoid inclination angle, and greater glenosphere overhang had a positive effect on postoperative fIR [47]. They reported that the effect was strongest with a 6 mm overhang. However, they are the only research group to date to investigate the effect of glenoid overhang in this context. Hochreiter et al. found a positive correlation between higher rates of scapular notching and good postoperative fIR [42]. This finding contradicts earlier studies, which indicated that lower rates of scapular notching are associated with better postoperative fIR [22,48]. To explain their results, the authors hypothesized that repeated external rotation during the postoperative period might enhance range of motion due to notching, thus improving fIR at long-term follow-up. This hypothesis, although not confirmed by our results, warrants serious consideration and further investigation in future studies based on existing literature. The findings of the present study hold clinical relevance as they contribute to the body of evidence regarding the impact of preoperative demographic, technical, and implant-associated factors on postoperative fIR following RTSA. This supports the need for continued research aimed at improving postoperative fIR outcomes.

This study has several limitations. First of all, the present study is a retrospective study, which inherits the biases of a retrospective design, such as the unavailability of preoperative functional and clinical outcome measures. However, to ensure comparability, strict inclusion and exclusion criteria were applied, including a monocentric inclusion, surgery with the same type of implant, using the same technique, and receiving the same standardized postoperative rehabilitation protocol. However, the strict inclusion criteria resulted in relatively small group sizes, which predisposes for a statistical type II error. Additionally, it should be noted that more arthroscopic than arthroplasty procedures are performed at the authors’ institute, which also explains the smaller sample size compared to other studies. As a result, an additional sampling bias, a reduced credibility, and an overfitting or an increased variability could have occurred. As this was a minimum two-year study, long-term results were no obtained, and therefore the long-term functional status might differ from the results presented here. Furthermore, humeral retrotorsion, scapular position was not evaluated when investigating possible factors influencing postoperative fIR. It should also be mentioned that the majority of patients (approximately 93%) received a non-lateralized glenosphere, meaning that, in this respect, no potential influence on fIR could be investigated.

## 5. Conclusions

None of the factors investigated, including implant-related parameters, demonstrated a significant influence on postoperative functional internal rotation (fIR) after reverse total shoulder arthroplasty (RTSA) in the studied cohort. This underscores the importance of considering factors such as humeral retroversion and scapular position and further suggests that postoperative fIR is a complex, integrated movement resulting from the interaction of various osseous and muscular structures of the shoulder girdle and the thorax. Consequently, it is unlikely that individual implant-related, demographic, or radiographic parameters exert a major influence on fIR outcomes. Further research should focus on a detailed analysis of the movement dynamics and interactions of the upper extremity both pre- and postoperatively, and on determining their potential impact on fIR following RTSA.

## Figures and Tables

**Figure 1 healthcare-12-01695-f001:**
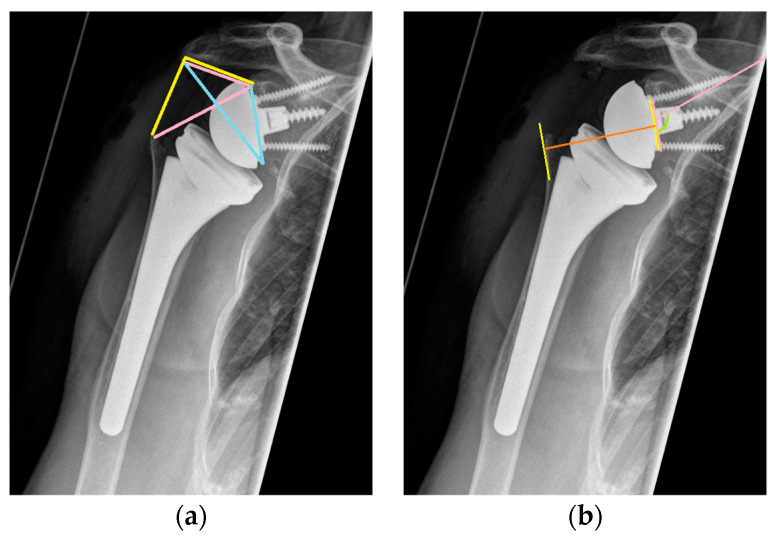
(**a**) yellow: lateralization shoulder angle (LSA); pink: distalization shoulder angle (DSA); blue: critical shoulder angle (CSA); (**b**) orange: lateralization of the glenoid baseplate; pink: inclination of the glenoid baseplate.

**Figure 2 healthcare-12-01695-f002:**
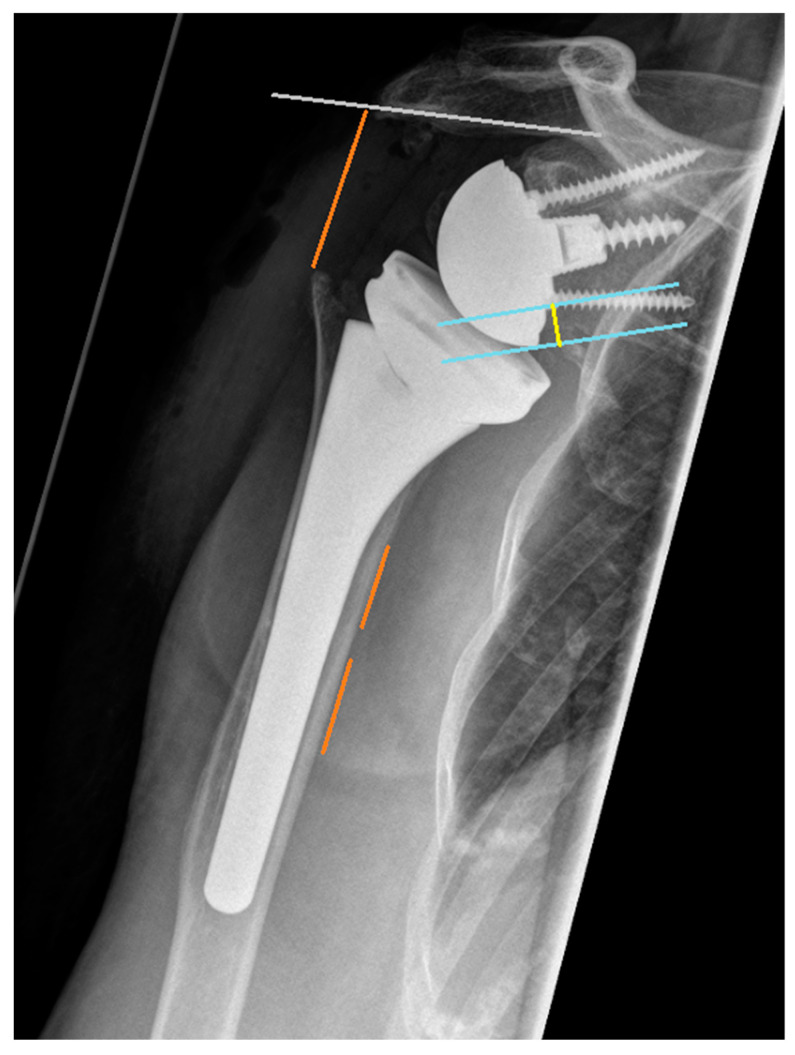
Orange: distalization of the greater tuberosity; yellow: glenoid overhang (GO).

**Figure 3 healthcare-12-01695-f003:**
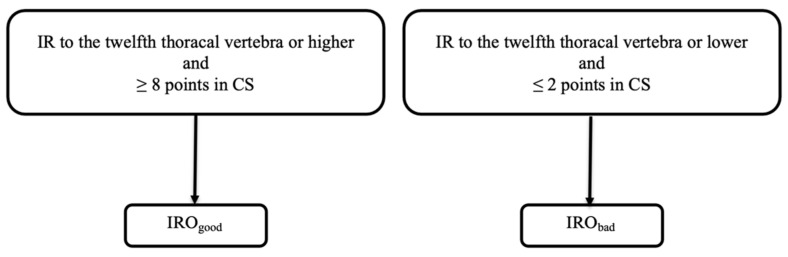
Flowchart displaying group allocation.

**Figure 4 healthcare-12-01695-f004:**
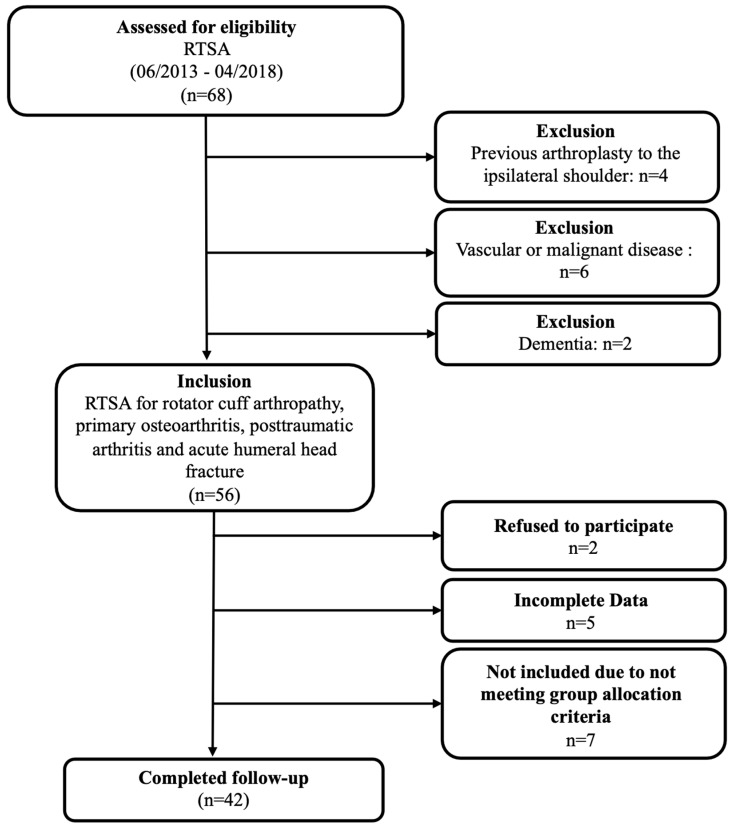
Flowchart displaying patients meeting study criteria.

**Table 1 healthcare-12-01695-t001:** Baseline characteristics.

	IRObad	IROgood	*p*-Value
Number of included patients, n	17	25	
Age at surgery (years)	73.7 ± 5	72 ± 6.1	0.337 n.s.
BMI (kg/m^2^)	28.7 (9.1) *	24.8 (5.6) *	0.086 n.s.
Follow-up (months)	42.2 (28) *	47 (35) *	0.356 n.s
Sex			
Male, n (%)	9 (52.9) *	11 (44) *	0.112 n.s.
Female, n (%)	8 (47.1) *	14 (56) *	0.059 n.s.
Postoperative Scores			
VAS Shoulder (points)	1 (0.5) *	1 (0) *	0.630 n.s.
SST (points)	8 (2.5) *	10 (4) *	0.037 n.s.
ASES (points)	85 (23) *	90 (73) *	0.076 n.s.
CS (points)	62.6 ± 11.6	70.8 ± 11.1	0.026 n.s.

Normally distributed continuous variables are shown as mean ± standard deviation. Non-normally distributed continuous variables, marked with an *, are shown as median (interquartile range), and categorical variables are shown as percentages. BMI, Body Mass Index; VAS, Visual Analogue Scale; SST, Simple Shoulder Test; ASES, American Shoulder and Elbow Surgeons Score; CS, Constant–Murley Score.

**Table 2 healthcare-12-01695-t002:** Functional assessment.

	IRObad	IROgood	*p*-Value
Postoperative Functional Assessment			
ROM active (°)			
Forward Flexion	130.6 ± 24.8	140.2 ± 18.5	0.158 n.s.
Abduction	120 (40.) *	120 (50) *	0.284 n.s.
External Rotation	27.4 ± 17	29.8 ± 18.3	0.662 n.s.
ROM passive (°)			
Forward Flexion	90 (0) *	90 (0) *	0.694 n.s.
Abduction	85 (10) *	90 (7.5) *	0.093 n.s.
External Rotation	28.2 ± 18.2	35.4 ± 21.2	0.262 n.s.
Abduction strength (N)	3.6 ± 2.1	3.7 ± 19	0.862 n.s.

Normally distributed continuous variables are shown as mean ± standard deviation. Non-normally distributed continuous variables, marked with an *, are shown as median (interquartile range). ROM, range of motion.

**Table 3 healthcare-12-01695-t003:** Radiographic assessment.

	IRObad	IROgood	*p*-Value
Radiographic Assessment			
Preoperatively			
LSA	98.2 (20.2) *	101.4 (8.8) *	0.969 n.s.
DSA	15.7 ± 5.8	16.5 ± 8.3	0.727 n.s.
CSA	31.8 ± 4.6	34.4 ± 5.6	0.118 n.s.
AHD	7.3 ± 4.7	6.4 ± 7.3	0.476 n.s.
Medialization	40.4 ± 10.3	41.8 ± 7.4	0.613 n.s.
Distalization	11.9 ± 6.3	14.1 ± 7.1	0.301 n.s.
Lateralization	49.7 (13) *	49.9 (13.9) *	0.505 n.s.
Glenoid inclination	102.4 ± 6.4	103.1 ± 7.3	0.746 n.s.
Glenoid version	83.4 (15.4) *	86 (7) *	0.595 n.s.
Glenoid overhang	3.4 ± 2.7	4.2 ± 2.9	0.373 n.s.
Postoperatively			
LSA	85.5 ± 8.1	88.9 ± 9.3	0.155 n.s.
DSA	46.7 ± 8	42.8 ± 11.2	0.214 n.s.
Medialization	18.7 ± 6.6	18.8 ± 19.8	0.969 n.s.
Distalization	36.9 ± 7	36 ± 6.6	0.343 n.s.
Lateralization	51.5 ± 5.8	52.4 ± 5.7	0.878 n.s.
Glenoid inclination	95.4 ± 9.3	96 ± 9.1	0.749 n.s.
Glenoid overhang	4.4 ± 3.4	5.8 ± 3.7	0.373 n.s.

Normally distributed continuous variables are shown as mean ± standard deviation. Non-normally distributed continuous variables, marked with an *, are shown as median (interquartile range). LSA, Lateralization Shoulder Angle; DSA, Distalization Shoulder Angle; CSA, Critical Shoulder Angle; AHD, Acromiohumeral distance.

## Data Availability

All data are available in a publicly accessible repository.

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
