# Peer review of "Demographic-, Radiographic-, and Surgery-Related Factors Do Not Affect Functional Internal Rotation Following Reverse Total Shoulder Arthroplasty: A Retrospective Comparative Study"

_healthcare, 2024, doi:10.3390/healthcare12171695_

Round 1

Reviewer 1 Report

Comments and Suggestions for Authors

The manuscript entitled "Evaluation of Clinical-Based and Radiographic Factors Affecting Functional Internal Rotation after Reverse Total Shoulder Arthroplasty: A Retrospective Comparative Study" aims to identify clinical-based and radiographic factors influencing postoperative fIR following RTSA. The study is well-structured, but several areas need significant improvement before it can be accepted for publication. Therefore, I recommend a Major Revision:

1. The manuscript needs a thorough review for clarity and flow. Several sections are difficult to follow due to convoluted sentence structures and technical jargon that is not adequately explained.

2. The introduction should provide a more comprehensive review of the existing literature. It should highlight the gap this study aims to fill.

3. The manuscript would benefit from clearly stated hypotheses. What specific outcomes were expected based on the literature review?

4. The choice of statistical tests should be justified, and the methods section should provide more detail about how assumptions for these tests were checked.

5. The abstract lacks detail about the statistical methods used and the specific results obtained. Please include more quantitative data in the abstract.

6. Provide a more detailed background on RTSA, including more recent studies. Also, clearly state the research question or hypothesis towards the end of the introduction.

7. Provide more details on the exclusion criteria and justify the exclusion of patients with vascular or malignant disease. Also, describe the surgical technique in greater detail, including any variations that might affect the outcomes. You also need to explain why the specific rehabilitation protocol was chosen and whether it was uniformly followed by all patients.

8. The results section should be reorganized for clarity. Present demographic data first, followed by clinical outcomes, and then radiographic outcomes. You can include more detailed tables and figures to summarize key findings. Ensure all tables and figures are referenced in the text.

9.  Discuss the implications of your findings in the context of existing literature. Also, address the limitations of your study in more detail, including potential biases and the impact of a relatively small sample size.

10. Provide more concrete suggestions for future research based on your findings.

11. Why were patients with vascular or malignant diseases excluded? How might these exclusion criteria impact the generalizability of your findings?

12. How were radiographic measurements standardized across different observers? Were inter-rater reliability tests performed?

13. Was the follow-up duration sufficient to capture the long-term outcomes of RTSA? What might be the potential effects of varying follow-up periods among patients?

14. How might the inclusion of subscapularis repair as a variable impact the outcomes of the study? Should this be considered more thoroughly in the analysis?

Comments on the Quality of English Language

Minor editing of the English language is required.

Reviewer 2 Report

Comments and Suggestions for Authors

This study addresses the clinical and radiological factors that impact functional internal rotation post-reverse total shoulder arthroplasty. This paper's introduction, research methods, conclusion, and review are well organized for the journal. However, providing this additional information will enhance reader comprehension.

1) According to the title, the study focuses on whether demographic and radiological parameters influence the deterioration of functional internal rotation after surgery. However, the conclusion that there are no influencing factors seems less persuasive to readers, so revising the title to reflect the study contents is necessary.

2) Abstract: Abbreviations such as VAS, SST, ASES, and CS  that are not defined in the research method of the abstract (Line 17) need to be modified, and a summary of the research method is required. For example, although it was stated that demographic and radiological parameters had no influence on study outcomes, a summary of these study variables was deemed necessary (Lines 12-25).

3) 2.7. The significance level in the '2.7 statistical analysis' was said to be 0.05, but in Table 1 (Line 194), the p-value column of the SST variable shows '0.037 n.s.'

4) Notes on abbreviations are required so Tables 1-3 can be read independently. 

Reviewer 3 Report

Comments and Suggestions for Authors

The manuscript "Evaluation of clinical-based and radiographic factors affecting functional internal rotation after reverse total shoulder arthroplasty: A retrospective comparative Study" ia aimed identify clinical-based and radiographic factors that influence postoperative functional internal rotation (fIR) following reverse total shoulder arthroplasty (RTSA). The work is well designed and very systematically done. The following suggestions are to be incorporated for enhancing the technical worthiness of the manuscript.

1. The introduction should be more clearly mention the technical gaps that has led to take up this work. 

2. Avoid group reference citation [Ex: [6,11,12,17,24-26].. Please mention the work of the researchers individually. 

3. The references should be more recent. There is only one reference from 2022. Most of the works referred are old.

4. Suggest to graphically represent the statistical data. 

Comments on the Quality of English Language

Good.

Round 2

Reviewer 1 Report

Comments and Suggestions for Authors

The authors have corrected everything. The manuscript is ready for publication.